# Return of Participants’ Incidental Genetic Research Findings: Experience from a Case-Control Study of Asthma in an American Indian Community

**DOI:** 10.3390/jpm13091407

**Published:** 2023-09-20

**Authors:** Lyle G. Best, Marcia O’Leary, Rae O’Leary, Wendy Lawrence, Dara G. Torgerson

**Affiliations:** 1School of Medicine and Health Sciences, University of North Dakota, Grand Forks, ND 58202, USA; 2Missouri Breaks Industries Research Inc., Eagle Butte, SD 57625, USA; marcia.oleary@mbiri.com (M.O.); rae.oleary@mbiri.com (R.O.); wendy.lawrence@mbiri.com (W.L.); 3Department of Epidemiology and Biostatistics, University of California San Francisco, San Francisco, CA 94158, USA; dara.torgerson@ucsf.edu

**Keywords:** genetics, return of results, American Indian, pediatrics, asthma

## Abstract

The proper communication of clinically actionable findings to participants of genetic research entails important ethical considerations, but has been challenging for a variety of reasons. We document an instance of the return of individual genetic results in the context of a very rural American Indian community, in hopes of providing insight to other investigators about potentially superior or inferior courses of action. This was a case/control study of asthma among 324 pediatric participants. Subsequently, microarray genotype data became available, providing over 2 million variants, incidentally including some conferring risk for conditions for which the American College of Medical Genetics recommends return of results. The study investigators engaged in extensive consultation with the IRB, the tribal government, and local clinicians to better inform our approach. We were able to notify the two participants heterozygous for the one clinically actionable variant identified. One participant welcomed this information and proceeded to obtain further clinical work-up; the other participant declined further follow-up. While demanding considerable time and effort, the return of clinically actionable genetic results is important from both an ethical perspective and to provide an improved trust relationship with the community of research participants.

## 1. Introduction

The return of individual, incidental genetic findings to research participants has been controversial [1,2] and uncommonly undertaken until recent movement in that direction [3,4]. Reluctance on the part of investigators to commit to this practice derives from a number of valid concerns, such as lack of Clinical Laboratory Improvement Amendments (CLIA)-certified results, frequent uncertainty as to the clinical implications of a particular genetic variant, lack of long-term grant funding to maintain contact with participants, and often insufficient support to provide adequate communication and counseling [5]. In counter-balance to these concerns is the inherent wish to provide potential benefit to individual participants who generously engage in research, a desire to prevent harm to a participant that is unaware of a potentially remediable genetic risk, and a respect for the autonomy of participants who wish to have information they feel may be useful.

It may be helpful to address occasional differences in terminology. Some (although not all [6]) view the terms “incidental” and “secondary” as applied to research findings as distinct, wherein “secondary” findings are considered those that are derived from analysis beyond the scope of the original study (such as analysis of a new topic conducted on an anonymized dataset) and “incidental” applies to results obtained in the course of the original, intended analysis, but not relevant to the research goal. The meaning of “incidental” in the current report complies with the latter, since the intent of this genotyping effort was solely to identify loci and potentially common variants that were associated with the asthma phenotype.

In routine clinical practice, it is common to encounter “incidental” information that was not sought in the diagnostic or treatment process. This information is routinely conveyed to the patient for consideration of additional diagnostic or treatment options. Indeed, failing to inform the patient of new information of a serious nature would be considered a breach of professional ethics. The American College of Medical Genetics has provided guidance in this area as it relates to the provision of clinical care.

The research setting is not the same as a clinical encounter, however, one might expect some overlap in the ethical evaluation of responsibilities. Certainly the research participant could be forgiven for assuming a certain degree of congruence, even though this issue may have been addressed in the consent. To the best of our knowledge, no research professional organizations have produced ethical standards to guide investigators on this issue. The eMERGE consortium, funded by the National Human Genome Research Institute, is an effort to utilize genome-scale screening to further both genetic research and direct patient care goals [7]. This naturally addresses issues inherent in the return of “incidental” genetic results.

In the current report, we report our experience in identifying clinically actionable incident genetic variant results among the MEGA^EX^ genotyping array (Illumina Inc., San Diego, CA, USA) results from a case–control study of genetic and environmental influences on the risk of asthma in an American Indian pediatric population [8]. We also describe the response of two participants to the incidental findings provided, and outline some of the difficulties inherent in this process.

## 2. Materials and Methods

Population-based study participants were ascertained through an unbiased query of the Indian Health Service (IHS) electronic medical records system for an array of diagnostic codes suggesting asthma. The factors influencing pediatric asthma (FIPA) study identified potential asthma cases and matched controls, from whom consent was requested for review of medical records to identify 108 cases and 215 controls that met study case and control criteria.

As with all research involving IHS facilities, FIPA was approved by the regional IHS institutional review board (IRB) and the tribal government, as well as the funding institution’s IRB. The consent forms advised that it was unlikely that any individual genetic test results would be returned, but if potentially useful results were developed, the study team would ask if they wished to be informed. The consent also stated that “…only genes that might be involved with asthma.” would be tested. To prevent any misinterpretation of what the consent stated, we present this section verbatim:

“In a study like this, what is found usually needs to be repeated by other researchers before it can be said “for sure” that something new is discovered. For these reasons you will probably NOT be contacted about results of your child’s genetic tests. If a new gene is found that would be important to predict your child’s risk for (or help your child avoid) asthma or other health problems we will contact you and ask, whether you would like to have the results of this gene testing explained to you”.

“This study will only test genes that might be involved in asthma; and no cells will be kept alive or cloned”.

Participants were told in lay terms that study genetic tests were not performed in Clinical Laboratory Improvements Amendment (CLIA)-approved facilities and that potential test results of clinical significance would probably need to be repeated in a “clinical lab” for confirmation. The consent also advised that payment for additional testing would not be available through the study. An explicit “opt-out” option for return of individual results was not provided in the consent.

After genotyping ten candidate variants individually and documenting some previously reported associations with asthma, we pursued genome-wide variant genotyping as a cost effective means to explore other potential genetic associations with asthma and asthma-related traits. Before utilizing a genotyping microarray and a genome-wide association approach, discussions as to whether and how to proceed were held with tribal members on the research team, local clinical providers, the IRB, and the tribal government. The tribal government approved the transfer of sufficient DNA to the University of Colorado for genotyping on the Illumina MEGA-EX microarray including ~2.3 million variants. Due to the large number of variants tested, it was anticipated that incidental findings of clinical significance could result. Incidental findings were determined from the American College of Medical Genetics (ACMG)-recommended clinically actionable variants for use in return of secondary results in the context of clinical diagnostic evaluation [9].

Plans for handling the possibility of returning clinically actionable results were developed and further consultations began with the same stakeholders as described above. Initially, it was suggested that community messaging inform both the general public and resident participants that FIPA now had extensive genetic results, and that there may be participants with clinically actionable results of importance. The rationale was that this would not stigmatize any individual due to direct contact from research staff for what could be presumed deleterious information; but that as participants came forward to receive confidential communication about their genetic results, most would be told there were no medically actionable variants to report. The participants would also be re-consented to accept this information prior to communicating their results.

However, our proposal to reach out to all study participants was viewed skeptically by most clinicians, tribal study staff, and the tribal government. The predominant suggestion was to simply contact affected individuals and ask if they wanted the genetic results returned, in line with expectations of what would occur in the case of an incidental finding in routine medical care. A major advantage of this approach is to avoid missing the very individuals most likely to benefit, if they either did not receive the community message or did not consider it sufficiently important to contact the research staff. In light of these consultations, it was decided to contact only those participants with clinically actionable results.

## 3. Results

Please see Figure 1 for a schematic outline of our approach to the challenge of returning individual, clinically actionable genetic results and the outcome of these efforts.

The demographic characteristics of the research participant community are important factors in fully understanding the challenges and benefits of implementing the return of incidental genetic results in this setting.

Study participants reside in an area of northern South Dakota covering 4266 square miles, with a population density between 2 and 3 people per square mile. Most tribal members live in cluster housing near small towns, on farmsteads, or in cluster sites far removed from basic services. Beyond tribal government and federally supported work in health care and education, ranching and farming provide the bulk of employment. Out of more than 3000 counties in America, two of the three counties in this community have the 2nd and 24th lowest per capita income [10], with 44% and 37% of residents having incomes below the poverty line [11]. More than 15% of adult residents have less than a high school education [11].

Although no “opt-out” option related to return of individual results was provided at the time of consent, the possible return of clinically relevant results was clearly stated in the consent and there were no potential parents or participants that indicated an unwillingness to enroll due to concern over this possibility.

The predominant source of medical care for tribal members in this community is the federally funded Indian Health Service and a tribal health department. The IHS hospital and clinic are typically staffed by three or four primary care physicians (often including a pediatrician) and an equal number of mid-level providers. The tribe operates an independent clinic with a couple of physicians and a similar number of mid-level providers. The closest specialty care is at least 90 miles and more commonly 170 miles distant (e.g., cardiac or pulmonary consultation). Although all recognized tribal members are eligible for care at the IHS facility, in order to be paid for with IHS funding, any non-IHS care must be pre-approved and is often deferred or delayed due to lack of funds or insufficient priority.

Both the community at large and individual participants were provided with information on the aggregate results of FIPA. This was accomplished by a newsletter to all participants including reference to a YouTube video created conveying our results, a radio broadcast, presentations to both the tribal council and local clinic staff, as well as tribal approval of each scientific publication.

All genetic data were de-identified during quality control and analysis as to protect the identity of study participants. Standard quality control for genome-wide association studies was performed, including the removal of variants with low call rates (<95%) and deviation from Hardy–Weinberg expectations. We also examined individual samples for unexpected genetic relatedness, low genotype call rates, and unexpected levels of heterozygosity (including heterozygosity of the X chromosome for consistency with reported gender). We then performed an intersection of variant identifiers with pathogenic variants listed by the ACMG, followed by visual confirmation of similar genome build, chromosome number, variant position, variant alleles (including strand), and genotype quality. We also compared overall allele frequency and heterozygosity among all FIPA participants for the identified ACMG pathogenic variants specifically, to provide an extra level of more stringent quality control without an arbitrary threshold. Overall, two individuals were identified as having clinically actionable genetic variants in the present genetic study. The two were siblings, and both lived 40 miles from the IHS facility and research headquarters.

The training and relationship to the community of the authors involved in returning these results is pertinent and briefly described here. The lead author (L.G.B.) is board-certified in family medicine, but with additional, informal training in both DNA diagnostic lab technique and genetic counseling during a year-long sabbatical at the Manitoba Health Sciences Center in Winnipeg, MB, Canada. His career with the IHS spanned over 20 years at a neighboring tribal community. In the subsequent 24 years, he has served as principal investigator on a number of genetic studies within American Indian communities. Three authors (M.O., W.L., and R.O.) are tribal members and/or married to a tribal member, live in the community, and have conducted various research efforts there for decades. The senior author (D.G.T.) is experienced in genetic studies and has collaborated with the lead author (L.G.B.) for close to 10 years. She has visited the community on several occasions to support collaborative research, participate in outreach activities, and provide genomics training.

Guided by the set of genes and variants the ACMG considers clinically actionable and pathogenic [12], a query of the MEGA-EX data found only a single *MYH7* coding variant, R870H (rs36211715) in two FIPA participants who were both heterozygous for this trait. The National Center for Biotechnology Information (NCBI) ClinVar database accession number (VCV000014120) from 2016 lists this variant as “pathogenic” as reviewed by an expert panel according to principles outlined by the “Recognition of Public Human Genetic Variant Databases” [13] of the US Food and Drug Administration.

The R870H variant is associated with hypertrophic cardiomyopathy (HCM) and the characterization as pathogenic is based on multiple clinical reports in the literature, from about 1995 as documented in dbSNP citations for rs36211715 [14]. Since this variant acts in a primarily dominant fashion (with penetrance of ~60%) [15], evidence of pathogenicity is derived mainly from intra-family linkage [16] and is at very low prevalence in the general population (frequency of the pathogenic allele is 2.3 × 10^−5^ in TOPMED [14]). The phenotype is characterized by otherwise unexplained left ventricular hypertrophy and increased risk of asymmetric septal hypertrophy and ventricular arrhythmias, occasionally leading to sudden cardiac death [15,16,17,18,19].

Prior to contacting participants, we identified funding to offer independent, CLIA-certified tests to confirm the presence of ACMG variants at no cost to the participant, since his does not have a mechanism to fund clinical follow-up of incidental genetic findings in a research setting. Following this, an attempt was made to contact the individuals by phone. Due to unstable cellular connections at the participants’ home and the COVID-19 pandemic, there was considerable difficulty making initial contact. However, one of us (L.G.B.) was able to visit with one of the siblings and their father for about 20 min and the preliminary results and their basic implications were explained. This author (L.G.B.) has over 30 years of direct clinical experience and provided genetic counseling services for a year under supervision at the Manitoba Health Sciences Center in Canada. During this session, the unconfirmed nature of our original results was emphasized, the general characteristics of the phenotype were outlined, and a release of information was obtained to review potentially pertinent medical records related to HCM. The participant, who was > 18 years of age at the time of contact, agreed to proceed with confirmatory testing. The Mayo Clinic was the most convenient reference genetic laboratory and arrangements were made to have a new blood sample collected at a local, private clinic. The original results were subsequently confirmed by the Mayo Clinic and the variant was interpreted as “pathogenic” according to their results summary.

After confirmatory results were available, a video teleconference was conducted with three of the authors to discuss the results and implications of this variant more fully. It was recommended (L.G.B.) that the participant have a cardiology consultation and 24 h ECG recording. Efforts to obtain referral from his for these services were initiated. Both an echocardiogram and the Holter monitor recording showed no current, anatomic evidence of HCM or arrhythmogenic concerns. Our final contact with this participant after this consultation was a written statement reiterating the findings from the cardiology consultation, the offer to provide additional testing for family members, the fact that the phenotype has limited penetrance, and strongly encouraging regular follow-up of blood pressure, repeat echocardiogram every few years, and general lifestyle recommendations for cardiovascular health. (see Appendix A). Contact information was also provided with encouragement to reconnect at any time with the study staff and physician.

Concurrent with above, multiple efforts to communicate further with this participant’s sibling were unsuccessful, including attempts by the engaged participant to reach out. A parent of the siblings was present during the initial 20 min phone consultation and did not decline offered testing personally, but has not attempted to follow-up and make arrangements for sample collection. The communicating participant has limited contact with the other parent and testing was not pursued. The full process, from the time the genetic variant was identified through our last contact with the family, lasted approximately 21 months.

## 4. Discussion

We present here a case study illustrating some of the difficulties in the return of incidental genetic findings in a research setting, and in a unique community context. Although consensus appears to be building that medically actionable research genetic findings should be conveyed to participants [7,20,21], many of the justified concerns of investigators and ethicists about implementation were encountered in the current study. This experience shows that American Indian research participants willingly consent to return of incidental genetic results, and that subsequent efforts to return clinically relevant incidental findings were eventually successful to the degree that one of two participants wished to engage. Some of these challenges are described below.

Participants’ desire to learn of individual genetic results from research participation is not universal, although the evidence accumulated to this point seems to indicate more acceptance than many anticipated [22,23,24,25,26]. Indeed, participants often reject their first choice to eschew return of results when given a second opportunity [1]. One of us (L.G.B.) with personal experience obtaining informed consent from over 800 research participants who were provided the option of allowing return of actionable results found only two opting out [27].

While investigators have often expressed support for return of aggregate study results, their actual performance has not been consistent or extensive [28,29]. In regard to return of individual research results, investigators have typically been more concerned about the desirability and challenges inherent in this process. Specific concerns have been related to return of results unauthorized by the participant, difficulties determining which variants are “clinically actionable”, potential long-term responsibilities of investigators, and possible lack of subsequent funding, as well as uncertainty about return of results to children [30,31,32,33].

The development of acceptable ethical standards and criteria for “clinically actionable” initially focused on whether and how best to return incidental findings uncovered during clinical investigation of conditions with a likely genetic origin. The increasing use of broad-based genetic testing, such as genotyping arrays and whole-genome sequencing, has resulted in a great number of unanticipated results, not directly associated with the primary diagnostic problem. The first professional body to provide guidance was the American College of Medical Genetics (ACMG) in 2013 [34]. This recommendation specifically listed 56 (currently 78) genes [35] with multiple, likely pathogenic variants that should be communicated to the individual via standard genetic counseling. A number of medical facilities have adopted congruent policies and research is in progress to assess the impact of return of genetic results in both a diagnostic realm and a more general screening approach [7,23,24,36].

Although many governmental consultative groups have suggested that similar standards should apply, at least in certain circumstances, to the return of results to research participants [6,20,37,38,39,40], to the best of our knowledge, no professional body has promulgated detailed standards similar to the ACMG [41]. Recently, the Global Alliance for Genomics and Health (GA4GH) promulgated an international policy on this subject [42], which we feel validates many of the principles applied in the current instance, particularly that of community engagement in the process.

Institutional review boards have, not surprisingly, taken a cautious approach to the problems associated with return of genetic research results [43]. Many have recognized the key role that IRBs should play in guiding appropriate return of results [20,21], and the valued principle of participant autonomy is often raised in these deliberations [44]. None the less, Wolf and Evans [31] raise serious concerns related to some recommendations to empower IRBs to set standards on an institutional or study-by-study basis. These concerns relate to the increased burden on often insufficient IRB resources, and often inconsistent interpretation of the ethical and regulatory context.

In the present study, the participant consents were permissive for the return of potentially useful information, including results related to “…other health problems” in addition to asthma, as stated in the consents. Considerable effort was also expended to garner input from the wider community of tribal leaders, local clinicians, and others as suggested by Raymond et al. [45]. We are grateful for this community-grounded input and feel that our initial implementation plans would likely have failed, at least to some extent. It is also fortunate that we had the benefit of supplemental research funds to allow CLIA-certified confirmation of our results, so this issue was easily resolved. Of note here, the release of laboratory results from a non-CLIA-approved facility is permitted under CLIA regulations when the testing was not performed for clinical purposes [31].

Although we encountered only two participants that met our qualifications for return of clinically useful results, just one of these (within the same family) was accepting of this information and utilized our recommendations. In spite of clearly informing one of the parents involved, there has been no further indication of interest in additional testing within the family. In this study, we were fortunate to identify a very small and manageable number of participants qualifying for return of results. Had the number identified been significantly larger, we may not have had the resources to follow our intentions. As it was, the challenges of geography, communications, health care financing, and sparse medical resources were substantial.

It is the authors’ strong conviction that return of clinically actionable results, whether genetic or not, is a moral obligation that cannot be overlooked, regardless of the many practical impediments it poses. In the United States, the watershed event precipitating a major reassessment of research ethics, the US Public Health Service Tuskegee Study, primarily failed in its duty to inform participants when treatment became available (i.e., “clinically actionable”). It is unfortunate that many in the biomedical research establishment continue to put more emphasis on practical impediments than on the underlying moral imperative.

While the experience of this study is limited in size and scope, we feel the unique community context provides useful additional information about the benefits and challenges related to what we feel is an extremely important ethical issue, namely, the return of clinically relevant genetic information to research participants.

## Figures and Tables

**Figure 1 jpm-13-01407-f001:**
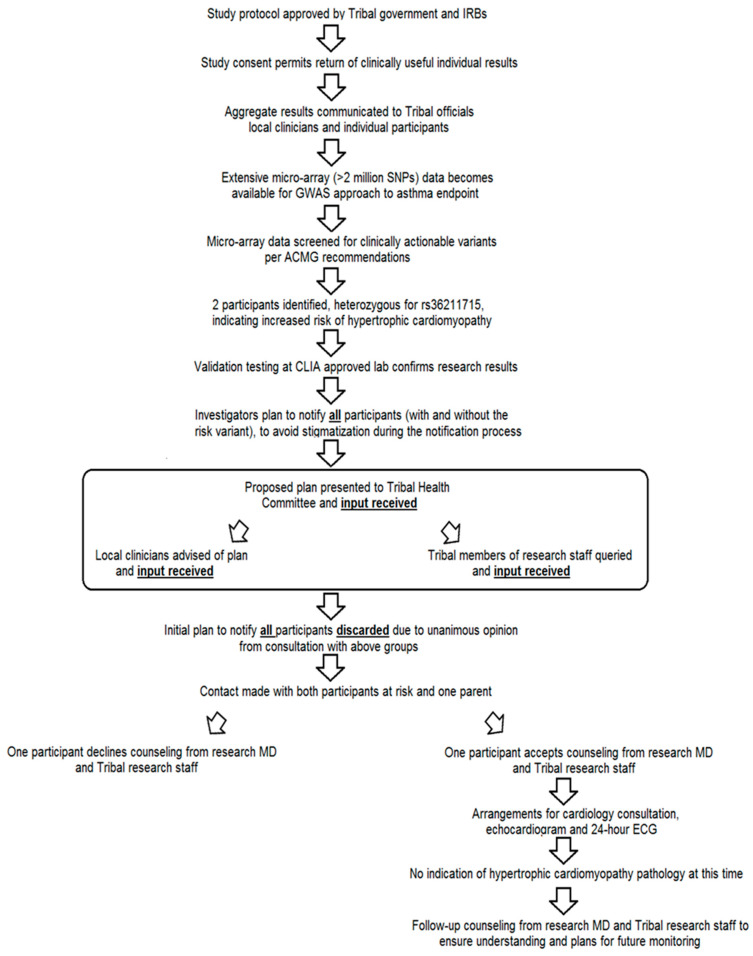
Schematic timeline of return of results.

## Data Availability

The data underlying the results of this study are owned and controlled by the Cheyenne River Sioux Tribe that approved its collection. This fact is clearly stated in the tribal resolution authorizing the research, and it must be recognized that this tribal community is an independent, sovereign government, in control over research activities within their borders. Access to data and materials is accomplished by application to Mr. Guthrie Duchneaux, IT director, Missouri Breaks Industries Research Inc., 505 S Willow St., Eagle Butte, SD 57625, 605-964-3419, email: guthrie.ducheneaux@mbiri.com, who will arrange for further consultation with the appropriate tribal official. Approximately 2 to 3 months may be required. The authors received special access privileges to the data due to their relationship with the tribal government, however, interested researchers who apply for data access will be able to access the same data as the authors.

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
