# Peer review of "Return of Participants’ Incidental Genetic Research Findings: Experience from a Case-Control Study of Asthma in an American Indian Community"

_jpm, 2023, doi:10.3390/jpm13091407_

Round 1
Reviewer 1 Report
The authors describe a logistically and ethically challenging topic. This is presented quite effectively by means of a detailed case example. I commend them for their thoughtful discussion of the topic. If permissible, it would be nice to include links to any information materials used as outreach as supplemental materials.
Reviewer 2 Report
The whole MS only described the results, which is not attracted to readers. This review recommends the authors present some figures for the results.
Reviewer 3 Report
This manuscript presented a case study of Asthma in an American Indian community, which overcome the challenges in participants’ return of results, and was able to notify the 2 participants heterozygous for the one clinically actionable variant identified. This investigation is helpful for community-based clinical study, which is usually limited to return of clinically relevant genetic information to research participants.
Reviewer 4 Report
Gist/sumamry: The authors come up with a gist of clinically actionable items on return of participants' incidental findings. They stress the need for terminologies, incidental/primary or secondary findings that are derived from the analyses beyond the scope.
They further setup a study based on CLIA guidelines and find the SNPs using a MEGA-EX microarray with a built-up of 2.3 M variants. It is assumed that the authors could find "incidental" findings from their work.
The work is a need of the hour and they have shown this on Indian registries associated with Asthma.
A pictorial methodology coalescing results will be very nice
What the author smsised was finding and explaining an extremely rare variant: https://www.ncbi.nlm.nih.gov/snp/?term=rs36211715 which they could have detailed whilst calling it as "incidental/pathogenic/extremely rare/pathogenic variants" etc
A line about ethics statement MJST also be mentioned in materials and methods
Scores on a scale of 0-5 with 5 being the best
Language: 4
Novelty: 4
Brevity: 3.5
Scope and relevance: 4
Minor but essential
counter balance could be used as one word
Results ARE pertinent
Round 2
Reviewer 2 Report
No comments.